# Predictive Factors for Nasogastric Tube Removal in Post-Stroke Patients

**DOI:** 10.3390/medicina59020368

**Published:** 2023-02-14

**Authors:** Shu-Ting Chuang, Ya-Hui Yen, Honda Hsu, Ming-Wei Lai, Yu-Fang Hung, Sen-Wei Tsai

**Affiliations:** 1Department of Nursing, Taichung Tzu Chi Hospital Buddhist Tzu Chi Medical Foundation, Taichung City 427213, Taiwan; 2Department of Nursing, Tzu Chi University of Science and Technology, Hualien 970046, Taiwan; 3Department of Nursing, National Chi Nan University, Puli Township 545301, Taiwan; 4Division of Plastic Surgery, Dalin Tzu Chi Hospital Buddhist Tzu Chi Medical Foundation, Chiayi City 622007, Taiwan; 5School of Medicine, Tzu Chi University, Hualien 970374, Taiwan; 6Department of Physical Medicine and Rehabilitation, Taichung Tzu Chi Hospital, Buddhist Tzu Chi Medical Foundation, Taichung City 427213, Taiwan; 7Department of Post-Acute Care Center, Taichung Tzu Chi Hospital Buddhist Tzu Chi Medical Foundation, Taichung City 427213, Taiwan

**Keywords:** stroke, nasogastric tube, rehabilitation, post-acute care

## Abstract

*Background and Objectives:* Stroke patients have different levels of functional impairment, and rehabilitation is essential to achieving functional recovery. Many post-stroke patients transition from acute treatment to post-acute care (PAC) with nasogastric tubes (NGTs) for rehabilitation. However, long-term NGT placement can lead to several complications, and its earlier removal can effectively reduce the incidence of mortality. This study aimed to use a PAC–cerebrovascular disease (CVD) program and physical functional evaluation scale tools to demonstrate the factors associated with NGT removal before post-stroke patient discharge. *Materials and Methods:* In this retrospective cohort study, data were collected between January 2017 and August 2022. We divided patients who had NGTs at admission into discharged with and without NGT groups to compare their baseline characteristics and physical functional status. Logistic regression analysis was used to detect the predictive factors for NGT removal. *Results:* There were 63 participants: 22 without NGT removal and 41 with NGT removal. The NGT removal rate was 65%. Age and scores for the activities of daily living by the Barthel index (BI), Functional Oral Intake Scale (FOIS), Mini-Mental State Examination, and Concise Chinese Aphasia Test were significantly different in terms of NGT removal status, but only the BI and FOIS were significantly correlated with NGT removal. Patients’ BI scores indicating severe to moderate dependence (21–90) had a 4.55 times greater chance of NGT removal (odds ratio, 4.55; *p* < 0.05) than patients who had total dependence (<20). Every one-point increase in FOIS score indicated a 3.07 times greater chance of NGT removal (odds ratio, 3.07; *p* < 0.05). *Conclusions:* The BI and FOIS evaluations may indicate the probability of NGT removal in patients.

## 1. Introduction

Stroke is the most common cause of death and disability worldwide and the second most common cause of death after cancer in Taiwan (Health Promotion Administration, MOHW, 2022) [1]. Most stroke patients have different levels of functional impairment, and rehabilitation is essential for patients to achieve functional recovery [2]. For post-stroke patients with physical functional deficits, a persistent rehabilitation program as a transitional phase between discharge and community return is required. The Taiwan National Health Insurance (TNHI) started a demonstration on improving the quality of post-acute care (PAC) in 2014. The PAC is a multidisciplinary care system that aims to increase patients’ functional status to facilitate their return to the community. Also known as PAC–cerebrovascular disease (PAC-CVD), it standardized a rigorous post-stroke inpatient rehabilitation program [3,4]. This PAC rehabilitation program is described as highly frequent and intense. Rehabilitation teams offer services in various disciplines based on patient abilities [5,6]. The maximum PAC-CVD unit stay was 12 weeks [7]. Many post-stroke patients transition from acute treatment to a PAC unit with a nasogastric tube (NGT). In clinical practice, NGT placement is used in approximately 10–15% of cases to maintain patient nutrition and hydration and avoid the risk of choking [8]. However, long-term NGT placement can carry many potential complications, such as mucosal damage ulcer, gastroesophageal reflux, and aspiration pneumonia [9]. Studies have also proven that NGT placement can affect the pharyngeal phase by causing an irregular epiglottis movement pattern [10]. Additionally, the NGT has been shown to contact the esophageal wall, possibly causing intramural esophageal bleeding [11]. These complications may further affect the prognosis of patients with stroke; therefore, earlier removal of the NGT can effectively reduce the incidence of mortality [12]. NGT insertion is uncomfortable for patients. Most post-stroke patients and their families are concerned about NGT removal. This study aimed to detect the predictors of NGT removal in a PAC-CVD unit and answer simple questions for patients about NGT removal. We used the PAC-CVD program’s physical functional evaluation scale tools to demonstrate the factors associated with NGT removal before discharge from the PAC-CVD program.

## 2. Materials and Methods

### 2.1. Patient Recruitment

This retrospective cohort study collected data from January 2017 to August 2022, and the post-stroke patients who were enrolled in this PAC-CVD plan were transferred to the PAC rehabilitation unit of Taichung Tzu Chi Hospital from acute care within 30 days after the onset of the first stroke. The first stroke was defined by the International Classification of Diseases 10th revision classification code (160, 161, and 163), and a Modified Rankin Scale functional score of 3–4. The total population comprised 351 patients. We included patients who had NGT at admission and excluded those with recurrent stroke, who were returned to acute care, who self-stopped the PAC program, or who died during their PAC-CVD unit stay. Thus, a total of 63 participants were included in the study. The participants were divided into discharge with NGT and discharge without NGT groups (Figure 1). The collected demographic and clinical variables included age, sex, stroke type (hemorrhagic or ischemic), length of stay, and physical functional evaluation scores.

### 2.2. Study Methods

The PAC-CVD program provided patients with highly intensive rehabilitation, including physical, occupational, and speech therapy. The schedule for daily rehabilitation from Monday through Friday, three times per day, for 50 to 60 min each. Depending on the patient’s daily state, a multidisciplinary rehabilitation team will alter the daily exercise routine. The maximum length of hospitalization for PAC-CVD patients is 12 weeks. Scaled evaluation tests for physical function were assessed upon admission to the PAC unit and at 3, 6, 9, and 12 weeks or at discharge from the PAC unit. Patients were divided into those discharged with NGT and those discharged without NGT. They all underwent scaled evaluation tests, including: (1) the Barthel daily living activity index (B-ADL: BI) for activities of daily living: score range, 0–100 [13]; (2) the Functional Oral Intake Scale (FOIS): score range, 1–7 [14]; (3) the Mini Nutritional Assessment (MNA): score range, 0–30 [15]; (4) the EuroQol Five Dimensions Questionnaire (EQ-5D) for evaluation of quality of life: score range, 1–3 [16]; (5) the Fugl-Meyer Motor Assessment Score (FMA): score range, modified sensation (0–44) and motor (0–66) [17]; (6) the Mini-Mental State Examination (MMSE): score range, 0–30 [18]; and (7) the Concise Chinese Aphasia Test (CCAT): score range, 1–12 [19]. There were some missing data among the EQ-5D responses; therefore, we used the variables of mobility and anxiety/depression in the EQ-5D for the statistical analysis. These outcome variables were used to determine the correlations and predictive factors for NGT removal.

### 2.3. Data Analysis

The patients’ baseline and demographic characteristics are presented as number (percentage) and mean (standard deviation). Pearson’s chi-squared test and the independent samples Mann–Whitney U test were used to compare the basic characteristics and outcome scores between the two groups. Variables that correlated with NGT removal in the univariate logistic regression model were further tested using a multivariable logistic regression model. The independent samples Mann–Whitney U test was used to compare the final outcomes of these two groups at admission and pre-discharge. Differences were considered statistically significant at *p* < 0.05. Analyses were performed using SPSS (version 13.0; SPSS Inc., Chicago, IL, USA).

### 2.4. Ethical Considerations

According to the TNHI guidelines, all post-stroke patients admitted to the PAC-CVD unit were required to sign an informed consent form authorizing the use of their anonymized medical data for research. This study was approved by the ethics committee of Taichung Tzu Chi Hospital (no. REC111-52).

## 3. Results

A total of 63 patients (28 female, 35 male) who underwent NGT placement were recruited from the PAC-CVD unit. The mean age was 68.8 ± 14.9 years, and the mean length of stay in the PAC-CVD unit was 60.6 ± 21.9 days. Sixteen patients (25.4%) had a hemorrhage stroke and 47 patients (74.6%) had an ischemic stroke. Overall, 41 patients (65%) underwent NGT removal before discharge, while 22 patients (35%) did not undergo NGT removal before discharge. The NGT removal rate was 65%. Four factors on the physical functional evaluation scales were statistically significant: BI, FOIS, MMSE, and CCAT. Age differed significantly between the two groups (Table 1 and Table 2). All variables were statistically analyzed using a univariate logistic regression model and further tested using a multivariate logistic regression model. There were two significant variables: severe/moderate dependence on BI (odds ratio, 4.55; *p* = 0.037) and FOIS (odds ratio, 3.07; *p* = 0.032) (Table 3). The significant variables of physical functional evaluation between these two groups at admission and discharge were as follows: BI (*p* < 0.001), FOIS (*p* < 0.001), MNA (*p* < 0.05), and anxiety/depression of EQ-5D (*p* < 0.05) (Table 4).

## 4. Discussion

This study aimed to identify factors associated with predicting NGT removal in post-stroke patients in a PAC-CVD unit. As shown in Table 1, we found that age, BI, FOIS, MMSE, and CCAT were significantly different between patients with versus without NGT removal. To further identify the predictive factors of NGT removal, the BI index scores were classified as follows: (1) total dependence, 0–20; (2) severe dependence, 21–60; (3) moderate dependence, 61–90; (4) slight dependence, 91–99; and (5) total independence, 100 [20]. The MMSE levels of impairment were also classified as: (1) severe cognitive impairment, 0–17; (2) mild cognitive impairment, 18–23; and (3) cognitively intact, 24–30 [21]. All variables were tested using a univariable logistic regression model and further tested using a multivariable logistic regression model, which showed that BI and FOIS were significant factors. In the BI index groups, most participants were at the level of total to moderate dependence, and none of the patients had slight dependence or total independence. Even patients with a BI level of severe or moderate dependence, with scores of 21–90, had a 4.55 times higher chance of NGT removal (*p* < 0.05) than those whose total dependence scores were <20. For each one-point increase in FOIS evaluation, patients had a 3.07 times higher chance (*p* < 0.05) of NGT removal. In the MMSE groups, this difference was not statistically significant.

Studies have reported that, compared to older patients, younger patients have a higher likelihood of NGT removal before discharge [22,23]. Another study also reported that age and functional independence scores for motor and cognitive abilities can predict oral feeding outcomes [24]. In our investigation, age, MMSE, and CCAT variables may have been correlated, but they were not statistically significant on the multivariable logistic regression and insufficiently powerful to predict NGT removal. Our findings also showed that NGT removal did not significantly correlate with stroke type (infarct or hemorrhage), a finding that is consistent with previous research findings [24,25]. In terms of the EQ-5D quality of life assessment, anxiety/depression was significantly different between the two groups. The NGT removal patients had a significantly improved mental state of depression/anxiety compared to those without NGT removal. A reported 31% of stroke patients experience depression, while 27% experience anxiety [26,27]. Post-stroke patients discharged with the NGT may require more psychological support and care. Previous studies reported that the NGT removal rate is approximately 40–58% in post-stroke patients [23,24]. In this study, the NGT removal rate was 65%, similar to our previous study [7]. Rehabilitation medical care provided by a multidisciplinary team can significantly improve patients’ physical and mental functions, emotions, nutrition, and mobility. Malnutrition is common in stroke patients due to a variety of factors such as physical discomfort, impaired body function, swallowing disorders, and social and psychological issues. Dysphagia affects 20–33% of stroke patients in the early stages. It often leads to complications in post-stroke patients, including nutritional deficiencies, dehydration, weight loss, decreased stamina, poor physical and mental recovery, pressure ulcers, poor wound healing, quality of life declines, and increased mortality [28]. An NGT is used to maintain basic nutritional needs. Eating orally can provide patients with more complete nutritional needs. The early prediction of NGT removal is not only a concern of patients and their families but also a treatment goal of the medical team [29].

Nutritional management of stroke patients is fundamental to positive rehabilitation outcomes. Therefore, early removal of the NGT can help patients obtain more complete nutrition. The MNA, BI index, and FOIS assessment results were significantly different between the two groups (Table 4). Early NGT removal improves patients’ FOIS and MNA assessments and further improves activities of daily living (ADL).

### Limitations

This retrospective study had some limitations. Our model of assessment variables may have lacked some factors. In addition, we were unable to determine whether patients whose NGT were removed remained in good condition after discharge. Further research exploring other potential associated factors and that ensures patients remain in good condition after NGT removal should be conducted.

## 5. Conclusions

This study showed that post-stroke patients’ age and physical functional assessment as measured by the ADL, FOIS, MMSE, and CCAT might impact the results of NGT removal. However, other unassessed factors may also have affected the results. ADL and FOIS improvement may be indicators of NGT removal and can answer patients’ questions about NGT removal. Post-stroke patients who are discharged with an NGT may require more psychological support and care after returning to the community. Importantly, NGT removal in the PAC-CVD unit was effective. The results of this study may serve as a guide for medical personnel looking to establish and develop more efficient rehabilitation delivery systems in PAC-CVD facilities.

## Figures and Tables

**Figure 1 medicina-59-00368-f001:**
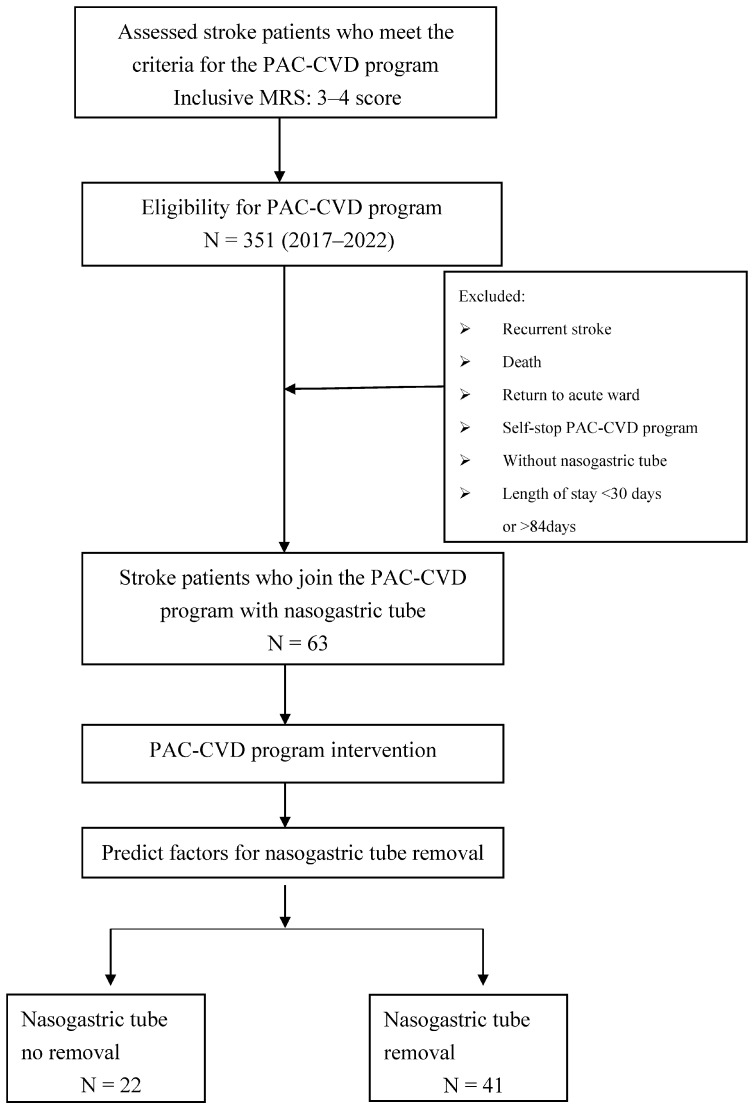
Study flow diagram. MRS, Modified Rankin Scale; PAC-CVD, post-acute care–cerebrovascular disease.

**Table 1 medicina-59-00368-t001:** Baseline characteristics of post-stroke patients.

	All	Nasogastric Tube Removed		
		No	Yes	
Variable	No. (%) Mean (SD) (*n* = 63)	No. (%) Mean (SD) (*n* = 22)	No. (%) Mean (SD) (*n* = 41)	*p* Value
Sex:				0.697
(1) Female	28 (44.4)	9 (40.9)	19 (46.3)	
(2) Male	35 (55.6)	13 (59.1)	22 (53.7)	
Age	68.8 ± 14.9	73.9 ± 13.5	66.1 ± 15.1	0.040
Stroke type:				0.382 *
(1) Hemorrhage	16 (25.4)	4 (18.2)	12 (29.3)	
(2) Ischemic	47 (74.6)	18 (81.8)	29 (70.7)	
Length of stay	60.6 ± 21.9	62.7 ± 23.5	59.5 ± 21.3	0.343
Evaluation index:				
1. BI (0–100)	19.7 ± 15.9	13.6 ± 12.9	22.9 ± 16.6	0.028
2. FOIS (1–7)	1.8 ± 1.3	1.3 ± 0.5	2.1 ± 1.5	0.005
3. MNA (0–30)	10.8 ± 4.9	11.1 ± 4.8	10.7 ± 4.9	0.675
4. EQ-5D:				
(1) Mobility (1–3)	2.3 ± 0.5	2.4 ± 0.5	2.2 ± 0.4	0.115
(2) Anxiety/depression (1–3)	1.8 ± 0.5	1.8 ± 0.4	1.8 ± 0.5	0.742
5. FMA:				
(1) Modified sensation (0–44)	24 ± 16.4	19.9 ± 16.3	26.2 ± 16.2	0.145
(2) Motor (0–66)	25.8 ± 23.5	21.7 ± 21.1	28 ± 24.6	0.216
6. MMSE (0–30)	14.0 ± 10.1	9.7 ± 10.4	16.3 ± 9.3	0.011
7. CCAT (1–12)	7.3 ± 3.8	5.7 ± 4.3	8.2 ± 3.3	0.025

* Fisher’s exact test. BI, Barthel index; CCAT, Concise Chinese Aphasia Test; EQ-5D, European quality of life index; FMA, Fugl-Meyer Assessment; FOIS, Functional Oral Intake Scale; MMSE, Mini-Mental State Examination; MNA, Mini Nutritional Assessment.

**Table 2 medicina-59-00368-t002:** Baseline data of the BI and MMSE groups.

	All Nasogastric Tube Removal				
			No	Yes	
Variable	Scores	No. (%) (*n* = 63)	No. (%) (*n* = 22)	No. (%) (*n* = 41)	*p* Value
BI groupings:					0.050
(1) Total dependence	0–20	42 (66.7)	19 (86.4)	23 (56.1)	
(2) Severe dependence	21–60	20 (31.7)	3 (13.6)	17 (41.5)	
(3) Moderate dependence	61–90	1 (1.6)	0	1 (2.4)	
(4) Slight dependence	91–99	0	0	0	
(5) Total independence	100	0	0	0	
MMSE groupings:					0.089
(1) Severe cognitive impairment	0–17	34 (54.0)	14 (63.6)	20 (48.8)	
(2) Mild cognitive impairment	18–23	13 (20.6)	6 (27.3)	7 (17.1)	
(3) Cognitively intact	24–30	16 (25.4)	2 (9.1)	14 (34.1)	

BI, Barthel index; MMSE, Mini-Mental State Examination.

**Table 3 medicina-59-00368-t003:** Predictive factors for nasogastric tube removal.

Variables	Crude OR	*p* Value	Adjusted OR (95% CI) ^†^	*p* Value
Sex (male vs. female)	0.80	0.679		
Age	0.96	0.053		
Stroke type (I vs. H)	0.54	0.339		
Length of stay	0.99	0.574		
Evaluation index				
1. BI:	1.05	0.034		
(1) Total dependence	Ref.		Ref.	
(2) Severe/moderate dependence	4.96	0.022	4.55 (1.10–18.90)	0.037
2. FOIS	3.38	0.019	3.07 (1.10–8.54)	0.032
3. MNA	0.98	0.729		
4. EQ-5D:				
(1) Mobility	0.41	0.117		
(2) Anxiety/depression	1.25	0.669		
5. FMA:				
(1) Modified sensation	1.02	0.146		
(2) Motor	1.01	0.304		
6. MMSE:	1.07	0.015		
(1) Severe cognitive impairment	Ref.			
(2) Mild cognitive impairment	0.82	0.758		
(3) Cognitively intact	4.90	0.056		
7. CCAT	1.19			

† Variables that are significantly correlated with NGT removal in the univariable logistic regression model were further tested in the multivariable logistic regression model. 95% CI, 95% confidence interval; BI, Barthel index; CCAT, Concise Chinese Aphasia Test; EQ-5D, European quality of life index; FMA, Fugl-Meyer Assessment; FOIS, Functional Oral Intake Scale; I vs. H, ischemic vs. hemorrhagic; MMSE, Mini-Mental State Examination; MNA, Mini Nutritional Assessment; NGT, nasogastric tube; OR, odds ratio.

**Table 4 medicina-59-00368-t004:** Comparison of variables at admission and pre-discharge.

	Nasogastric Tube Removal		
Variables	No Mean (SD) (*n* = 22)	Yes Mean (SD) (*n* = 41)	*p* Value
1. BI (0–100) diff.	20.7 ± 19.9	44.5 ± 17.1	<0.001
2. FOIS (1–7) diff.	1.1 ± 1.5	4.2 ± 1.7	<0.001
3. MNA (0.0–30.0) diff.	1.1 ± 2.3	2.2 ± 2.4	0.027
4. EQ-5D:			
(1) Mobility (1–3) diff.	−0.2 ± 0.6	−0.4 ± 0.5	0.127
(2) Anxiety/depression (1–3) diff.	−0.2 ± 0.5	−0.6 ± 0.6	0.018
5. FMA:			
(1) Modified sensation (0–44) diff.	8.9 ± 12.9	5.9 ± 8.8	0.417
(2) Motor (0–66) diff.	12.8 ± 16.7	14 ± 12.6	0.572
6. MMSE (0–30) diff.	4.3 ± 4.6	4.7 ± 4.3	0.434
7. CCAT (1–12) diff.	0.5 ± 2.9	1.4 ± 1.5	0.275

BI, Barthel Index; CCAT, Concise Chinese Aphasia Test; diff., pre-discharge-admission; EQ-5D, European quality of life index; FMA, Fugl-Meyer Assessment; FOIS, Functional Oral Intake Scale; MMSE, Mini-Mental State Examination; MNA, Mini Nutritional Assessment.

## Data Availability

The data presented in this study are available upon request from the corresponding author.

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
