# Peer review of "Predictive Factors for Nasogastric Tube Removal in Post-Stroke Patients"

_medicina, 2023, doi:10.3390/medicina59020368_

Round 1
Reviewer 1 Report
This essay concerning CVA-PAC program and the predicting factors of NG removal during PAC in Taiwan. Here are some of my questions.
1. BI score is significantly correlated to NG removal. The authors should elaborate more about the each subunit of BI (maybe by a table or figure?) and their correlations. For example, BI score includes feeding and the patients' ability to self-feeding, authors can explain if the "feeding" score in BI is correlated to NG removal and may give reason showing why improvements in fine motor correlate NG removal.
2. The relationship between MNA score and NG removal could by explained more in the discussion.
3. Did all these patients with NG receive speech and swallowing therapy? If yes, the how often did the therapy deliver? Authors could analysis how many patients in each group(with/without NG) receive speech and swallowing therapy(by table or figure) and explain if swallowing therapy contribute to NG removal
Thank you!
Reviewer 2 Report
This is a lovely article, few points can be considered.
1. Barthel index is a generalized score not for NGT
2. MNA Mini Nutrition Assement
3. Fugl Meyer Assessment
these are how to corelate with outcome of your conclusion
Reviewer 3 Report
Abstract section:
With the findings of the study, this inference cannot be made in the conclusion part, which starts with line 38. “The PAC-CVD program could 38 provide post-stroke patients with high-quality rehabilitation treatment.”
Introduction section:
Detailed information about the PAC-CVD program is given. But does this program include swallowing rehabilitation? Is there any special treatment to speed tube removal in patients with NGT?
Why is it important to predict NGT removal? What would be the advantages of being able to predict this? You need to explain this a little more.
Method section:
In NGT removal; Was NGT removed on the day the patients were discharged? Or was it removed at any time during his stay in the hospital? ; Do you have data on exactly when the NGT was removed?
I think the direction of the arrow in the "excluded" box on line 100 is reversed. It needs to be fixed.
Information such as CCAT, MMSE are given in both 2.1 and 2.2. It can be removed from 2.1 so that it does not happen again.
Results section:
There are two "Table 1". Table numbers should be corrected.
"table 2" starting with line 170 is not fully understood. The most important table of the study should be rearranged.
Discussion section:
The importance of predicting the removal of NGT has not been fully emphasized and discussed. The inference in the sentence starting with line 215 cannot be reached according to the results of this study. This sentence should be removed.
Round 2
Reviewer 1 Report
Please describe the frequency and the duration of physical, occupational, speech/swallowing therapy the patients all receiving in the "Method" section.
Thanks!
